Corrected: Author correction

# Nitrogen-rich organic soils under warm well-drained conditions are global nitrous oxide emission hotspots

Jaan Pärn [1,2,3], Jos T.A. Verhoeven[4], Klaus Butterbach-Bahl[5], Nancy B. Dise[6], Sami Ullah[3], Anto Aasa[1], Sergey Egorov[1], Mikk Espenberg[1], Järvi Järveoja[1,7], Jyrki Jauhiainen[8], Kuno Kasak[1], Leif Klemedtsson[9], Ain Kull[1], Fatima Laggoun-Défarge[10], Elena D. Lapshina[11], Annalea Lohila[12], Krista Lõhmus[13], Martin Maddison[1], William J. Mitsch[14], Christoph Müller[15,16], Ülo Niinemets[17], Bruce Osborne[16], Taavi Pae[1], Jüri-Ott Salm[18], Fotis Sgouridis [19], Kristina Sohar[1], Kaido Soosaar[1], Kathryn Storey[20], Alar Teemusk[1], Moses M. Tenywa[21], Julien Tournebize[22], Jaak Truu[1], Gert Veber[1], Jorge A. Villa [23], Seint Sann Zaw[24] & Ülo Mander[1]

Nitrous oxide ($N_2O$) is a powerful greenhouse gas and the main driver of stratospheric ozone depletion. Since soils are the largest source of $N_2O$, predicting soil response to changes in climate or land use is central to understanding and managing $N_2O$. Here we find that $N_2O$ flux can be predicted by models incorporating soil nitrate concentration ($NO_3^-$), water content and temperature using a global field survey of $N_2O$ emissions and potential driving factors across a wide range of organic soils. $N_2O$ emissions increase with $NO_3^-$ and follow a bell-shaped distribution with water content. Combining the two functions explains 72% of $N_2O$ emission from all organic soils. Above 5 mg $NO_3^-$-N kg$^{-1}$, either draining wet soils or irrigating well-drained soils increases $N_2O$ emission by orders of magnitude. As soil temperature together with $NO_3^-$ explains 69% of $N_2O$ emission, tropical wetlands should be a priority for $N_2O$ management.

[1] Department of Geography, Instute of Ecology and Earth Sciences, University of Tartu, Tartu 51014, Estonia. [2] School of Geography, Geology and the Environment, Keele University, Newcastle ST5 5BG, UK. [3] School of Geography, Earth and Environmental Sciences, University of Birmingham, Birmingham B15 2TT, UK. [4] Ecology and Biodiversity, Department of Biology, Utrecht University, Utrecht 3584 CH, The Netherlands. [5] Institute of Meteorology and Climate Research, Karlsruhe Institute of Technology, Garmisch-Partenkirchen 82467, Germany. [6] Centre for Ecology and Hydrology, Edinburgh EH26 0QB, UK. [7] Department of Forest Ecology and Management, Swedish University of Agricultural Sciences, Umeå SE901 83, Sweden. [8] Natural Resources Institute Finland (Luke), Helsinki FIN-00790, Finland. [9] Department of Earth Sciences, University of Gothenburg, Gothenburg SE405 30, Sweden. [10] Institute of Earth Sciences, National Center for Scientific Research (CNRS) and University of Orléans, Orléans 45100, France. [11] UNESCO Chair of Environmental Dynamics and Climate Change, Yugra State University, Khanty-Mansiysk 628012, Russia. [12] Atmospheric Composition Research, Finnish Meteorological Institute, Helsinki FIN-00101, Finland. [13] Department of Botany, Institute of Ecology and Earth Sciences, University of Tartu, Tartu 51014, Estonia. [14] Everglades Wetland Research Park, Kapnick Center, Florida Gulf Coast University, Naples 4940 FL, USA. [15] Institute of Plant Ecology, Justus Liebig University Giessen, Giessen 35392, Germany. [16] University College Dublin (UCD) School of Biology and Environmental Science, UCD Earth Institute, Dublin 4, Ireland. [17] Department of Plant Physiology, Institute of Agricultural and Environmental Sciences, Estonian University of Life Sciences, Tartu 51014, Estonia. [18] Estonian Fund for Nature, Tartu 51014, Estonia. [19] School of Geographical Sciences, University of Bristol, Bristol BS8 1SS, UK. [20] Department of Primary Industries, Parks, Water and Environment, Tasmanian Government, Hobart 7001 TAS, Australia. [21] Department of Agricultural Production, College of Agricultural and Environmental Sciences, Makerere University, Kampala 7062, Uganda. [22] Hydrosystems and Bioprocesses Research Unit, National Research Institute of Science and Technology for Environment and Agriculture (IRSTEA), Antony 92160, France. [23] Grupo de Investigación Aplicada al Medio Ambiente, Corporacion Universitaria Lasallista, Caldas 51 118, Colombia. [24] Forest Resource Environment Development and Conservation Association, Yangon 0951, Myanmar. Correspondence and requests for materials should be addressed to J.Pär. (email: jaan.parn@ut.ee)

Organic soils make up more than one-tenth of the world's soil nitrogen (N) pool[1] and are a significant global source of the greenhouse gas nitrous oxide ($N_2O$). We do not fully understand the underlying microbial production and consumption processes and how these interact with environmental drivers such as the microclimate, physics, and chemistry of the soil[2]. $N_2O$ can be emitted as a by-produce of both incomplete nitrification and incomplete denitrification. Under anaerobic conditions, N is primarily conserved in organic compounds, and nitrification (the conversion of ammonium ($NH_4^+$) to $NO_3^-$) is limited to the rooting zone or is absent. The normally low availability of $NO_3^-$ also restricts rates of denitrification (the conversion of $NO_3^-$ to $N_2$) in anaerobic soil; if sufficient $NO_3^-$ is present but oxygen remains restricted, denitrification may go to completion, producing atmospheric $N_2$[3–6]. Reduction of soil moisture promotes mineralisation of organic N to $NH_4^+$, which can be nitrified to $NO_3^-$[7,8], and produces the partially-oxidised conditions that are conducive to incomplete denitrification, a major source of $N_2O$[9]. $N_2O$ emission has been both positively and negatively correlated with soil moisture, as water-filled pore space (WFPS) or volumetric water content (VWC)[10–26] depending upon water status: intermediate levels of around 50–80% WFPS or VWC appear to be optimal for $N_2O$ production[26–36].

Increases in soil temperature normally enhance $N_2O$ production[37] up to about 24 °C, where bacterial denitrification reaches an optimum[38,39], above which $N_2O$ efflux drops. However, denitrifier communities may adapt to higher temperatures, leading to further increases in $N_2O$ emissions[40]. A review of laboratory and field studies shows inconsistent relationships between temperature and $N_2O$ emissions[13,21,41] from strongly positive to negative, illustrating that temperature alone cannot explain $N_2O$ fluxes but must be considered in the context of other drivers, especially soil moisture. At near-zero soil temperatures, the freeze-thaw effect may produce significant amounts of $N_2O$[42–45].

As growing population pressure has increased the extent of fertilised and drained organic soil, nitrogen-rich organic soils will become increasingly important global $N_2O$ sources[2,46]. Currently $N_2O$ contributes 12% of $CO_2$-equivalent GHG emissions from land use in tropics[47]. Quantifying the influence of both increasing rates of land drainage and climate change on organic soil $N_2O$ fluxes is thus critically important[2]. However, emission factors used to assess $N_2O$ fluxes from different land uses and ecosystems are usually simple proportions of the application rate of fertiliser (or atmospheric reactive N deposition for non-cultivated soils) and broad land-use categories; these models also do not take into account climate-related changes[48]. Thus we lack an inclusive model to quantify the potential of organic soils to be a globally important source of $N_2O$[2,49]. To address this challenge we undertook a standardised global survey of in situ $N_2O$ fluxes from organic soils, together with ancillary measurements of key drivers, to derive a model of $N_2O$ emissions that would be applicable to a wide range of biomes and environmental conditions. We find that $N_2O$ emission from organic soils increases with rising soil $NO_3^-$, follows a bell-shaped distribution with soil moisture, and increases with rising soil temperature. This emphasises the importance of warm drained fertile soils to climate change.

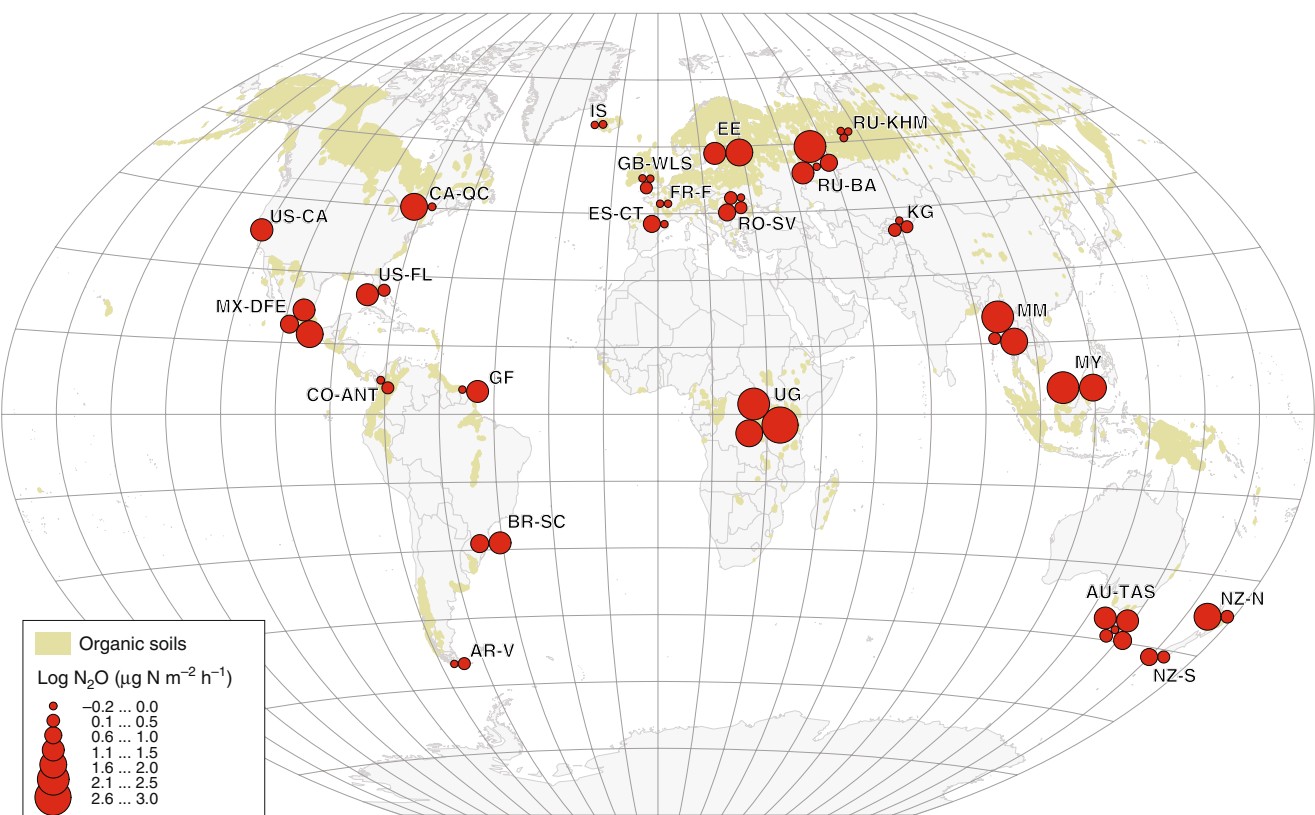

**Fig. 1** Site-mean $N_2O$ fluxes by study region superimposed on a global organic-soil map. Country and region codes are defined after ISO 3166-2. The distribution of organic soil was defined as >150 t $C_{org}$ ha$^{-1}$ from the Global Soil Organic Carbon Estimates (courtesy of the European Soil Data Centre) + 0.5 geographical-degrees buffer for visual generalisation

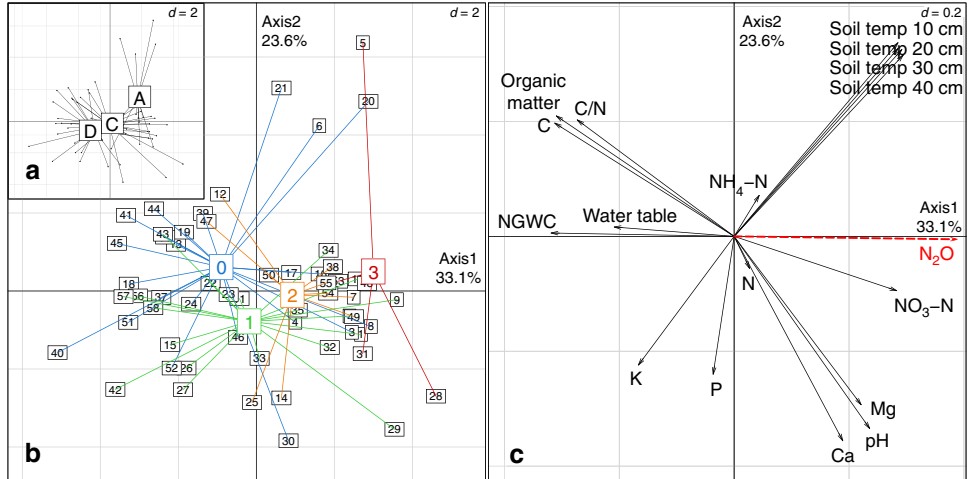

**Fig. 2** Ordination plots based on principal component analysis grouping sites and variables. **a** Köppen climates (A) tropical, (C) temperate and (D) boreal; **b** intensity of agricultural use (0) no agriculture, (1) moderate grazing or mowing, (2) intensive grazing or mowing and (3) arable; **c** soil physical and chemical parameters. $N_2O$ emission used as passive variable. *d*: grid scale, VWC volumetric water content. See Supplementary Data 1 for site names

## Results

**Principal component analysis**. Site-mean $N_2O$ fluxes by study region superimposed on a global organic-soil map are shown in Fig. 1. The principal component analysis (PCA) differentiated tropical sites from temperate and boreal ones, and low agricultural-intensity sites (index 0 and 1) from arable sites (index 3) ($p < 0.05$; Fig. 2a, b). Soil $NO_3^-$ was positively related to $N_2O$ emission; VWC and water table were strongly negatively correlated with $N_2O$ emissions, and C/N, C, and organic matter were less strongly negatively correlated with $N_2O$ emissions, and soil temperature was positively related to $N_2O$ emissions (Fig. 2c). Soil-available P was orthogonal to the $N_2O$-flux vectors, indicating no correlation. The difference between $N_2O$ emissions from drained and natural sites was clear in all three major climate types (Supplementary Table 1).

**Global models**. Of the 18 parameters assessed (Supplementary Data 1), soil $NO_3^-$ was the strongest predictor of site-mean $N_2O$, explaining 60% of the variation in log $N_2O$ flux (Fig. 3a). The generalised additive model (GAM) trend was similar to concave log-log quadratic. Inclusion of site-mean VWC (Fig. 3b) raised the explanatory power of the multiple-regression GAM to 72% ($n = 58$; $R^2 = 0.72$; $p < 0.001$; Eq. (1); Fig. 4a). The regression surface was similar to a convex paraboloid with an apex at approximately 50% VWC:

$$\begin{aligned}\mathrm{Log}(N_2O{-}N + 1) &= 0.035 + 0.39\,\mathrm{log}NO_3{-}N \\ &\quad + 0.025\left(\mathrm{log}NO_3{-}N\right)^2 \\ &\quad + 4.8\,\mathrm{VWC} - 5.2\,\mathrm{VWC}^2\end{aligned} \quad (1)$$

In an independent comparison of the model with published data, relative $N_2O$ emissions were represented well. The relationship between the mean $N_2O$ fluxes (relative to the maximum value in the respective external data set) and VWC was best described by a bell-shaped GAM regression curve ($R^2 = 0.78$; $p < 0.001$; Fig. 5) similar to the VWC component of our global model (Fig. 3b). Both curves peaked at around 50% soil moisture.

Both our model and the literature support the idea that fluctuation around the intermediate VWC ($\sim 0.5\,\mathrm{m}^3\,\mathrm{m}^{-3}$) creates variability in the oxygen content within the soil profile. That, in turn, stimulates mineralisation and nitrification which contribute to higher $NO_3^-$ content[8,9,50,51]. Intermediate VWC also promotes incomplete denitrification, in agreement with early conceptualisations[25,40], previous regional-scale studies[28,33–35] and experiments[9,51,52]. The maximum $N_2O$ emission at the intermediate VWC means that both wetting from lower moisture values and drying from higher moisture will increase $N_2O$ emissions. At a VWC of $\sim 0.8\,\mathrm{m}^3\,\mathrm{m}^{-3}$, oxygen concentration in the pore water is 5–9% of saturation, which is low enough to trigger $N_2O$ production but insufficient for complete denitrification[9,51,52].

There was no significant relationship between $N_2O$ flux and $NH_4$-N among our observations ($p = 0.79$), suggesting that denitrification was probably the main source of $N_2O$ emissions rather than nitrification. Only one site (Tasmania drained fen 2) directly received mineral fertiliser, whereas the nitrate in the other 57 sites originated from livestock and natural sources such as nitrification, atmospheric deposition, runoff and groundwater. Thus our global model describes $N_2O$ emission due to grazing and naturally transported nitrate.

We found only a weak relationship between $N_2O$ fluxes and soil temperature (40cm-depth temperature log GAM $R^2 = 0.21$, $p < 0.001$; Fig. 3c). The soil temperatures normalised to local annual air-temperature maxima gave even lower correlation values (e.g. with temperature at 40 cm-depth log GAM $R^2 = 0.09$, $p = 0.018$). This may have been partially due to the short time span of our measurements per site. However, that is consistent with the meta-analysis of published data in eleven papers showing no correlation between long-term $N_2O$ fluxes and soil temperature[17,24,31,32,34,53–58]. The test for an upper boundary[59] in our temperature data was negative ($p > 0.05$). Therefore we accepted the $H_0$ hypothesis that our data are from a bivariate normal process and so the envelope of the data points does not represent a boundary. This also suggests that the high $N_2O$ fluxes were measured in soils where temperature was not the limiting factor. A multiple-regression GAM model containing soil temperature at 40 cm depth and log $NO_3^-$

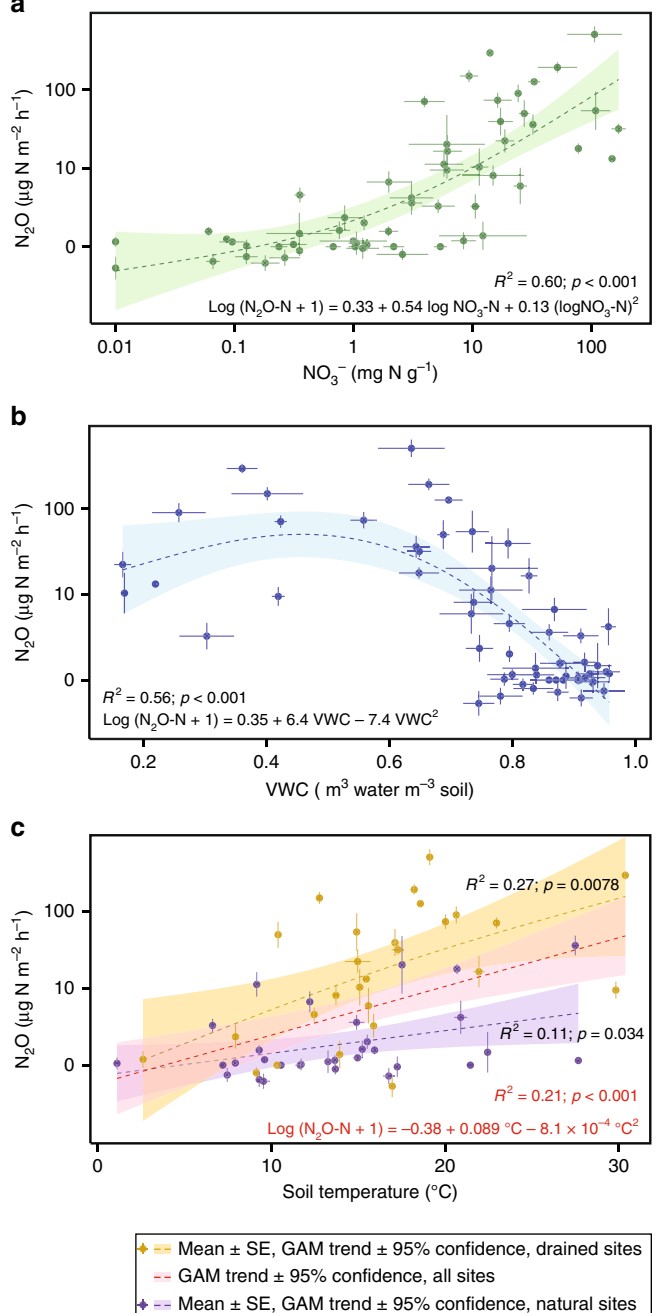

**Fig. 3** Relationships between site-mean $N_2O$ fluxes and soil parameters. The panels correspond to the relationships between $N_2O$ fluxes and: **a** nitrate; **b** volumetric water content; **c** soil temperature at 40 cm depth across all sites, and drained and natural sites. The error bars correspond to standard errors of the mean (s.e.m.). $N = 58$

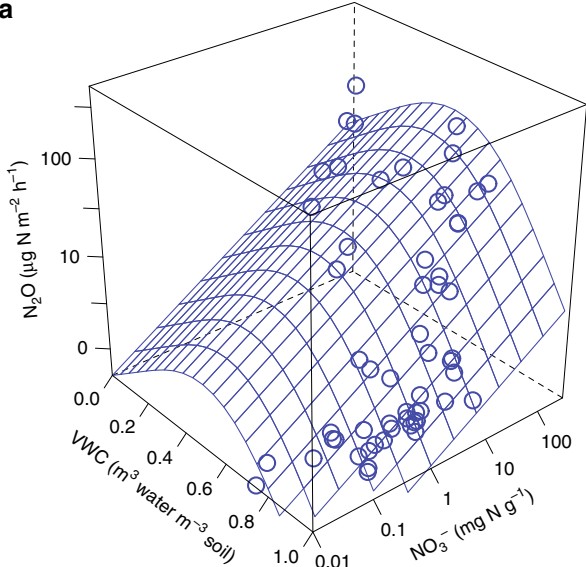

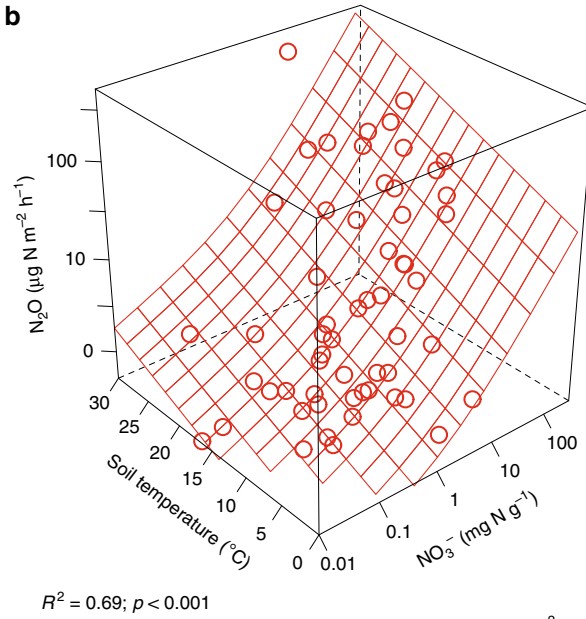

**Fig. 4** Site-mean $N_2O$ flux multiple-regression models. **a** Soil nitrate and volumetric water content (VWC); **b** soil nitrate and temperature at 40 cm depth. $N = 58$

explained 69% of log $N_2O$ fluxes ($n = 58$; $R^2 = 0.69$; $p < 0.001$; Eq. (2); Fig. 4b):

$$\text{Log}(N_2O-N+1) = -0.15 - 0.50\,\text{logNO}_3-N \\ + 0.10\,(\text{logNO}_3-N)^2 \quad (2) \\ + 0.036\,^\circ C + 1.9 \times 10^{-5}\,^\circ C^2$$

Within our drained sites (Supplementary Data 1; $n = 27$) the temperature relationship was somewhat stronger ($R^2 = 0.27$; $p <$

0.0078; Fig. 3c). This shows that organic soils exposed to warmer conditions, such as in the tropics, can act as $N_2O$-emission hotspots where soil moisture is optimal (Fig. 3b) and $NO_3^-$ is above a threshold of 5 mg N $kg^{-1}$ (Fig. 3a).

Because we sampled each site for only a few days and that we visited temperate and boreal sites during the growing seasons this study was not designed to detect the effect of seasonal or synoptic-scale variation of temperature, soil nitrate, and other factors within each site. Thus our global models are only applicable to estimate daily $N_2O$ emissions based on instantaneous environmental conditions at organic-soil sites. Annual-average $N_2O$ emissions at sites under a seasonal climate may be

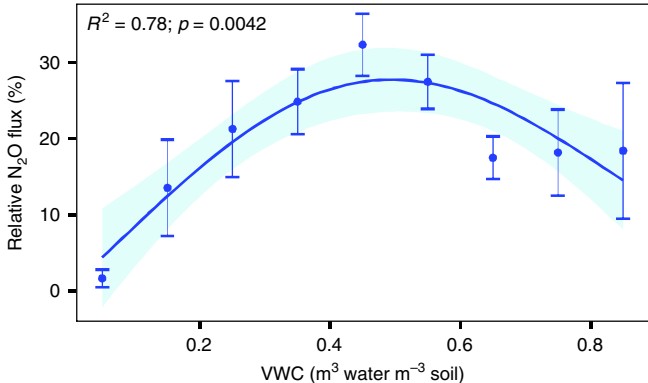

**Fig. 5** Relative $N_2O$ fluxes versus volumetric water content (VWC) in 11 published annual time series. The $N_2O$ fluxes are scaled to the maximum value measured at each respective site. The dots and whiskers are average ± s.e.m. within the respective soil-moisture class. The curve is the GAM regression ($k = 3$) between average relative $N_2O$ fluxes and VWC. The light blue area marks the 95% confidence limits of the regression line

more difficult to draw from our model. Yet the model could be useful to estimate $N_2O$ emissions at sites under a lack of seasonal variation in environmental conditions such as the humid tropical climate. Upscaling our three tropical sites with intensive land use (the Malaysian oil palm plantation, and the Myanmar and Uganda arable sites; Supplementary Data 1) to a year's duration and comparing them with the special default emission values (EF2) in IPCC Guidelines 2006 for tropical organic soils[60] (16 kg $N_2O$-N ha$^{-1}$ y$^{-1}$, range 5–48 kg ha$^{-1}$ y$^{-1}$) gave us the following results. For the Malaysian site, soil temperature was 27–30 °C, the mean emission rate was 294.3 µg $N_2O$-N m$^{-2}$ h$^{-1}$ = 25.8 kg $N_2O$-N ha$^{-1}$ y$^{-1}$. For Myanmar, 14–19 °C (upland), the figures were 125.5 µg $N_2O$-N m$^{-2}$ h$^{-1}$ = 11.0 kg $N_2O$-N ha$^{-1}$ y$^{-1}$. For Uganda, 17–20 °C (upland), the figures were 507.3 µg $N_2O$-N m$^{-2}$ h$^{-1}$ = 44.5 kg ha$^{-1}$ y$^{-1}$. Thus the annual fluxes obtained by this simple upscaling all fell within the IPCC tropical default range.

**Other potential drivers**. The logarithm of C:N ratio, a common scalar explanatory variable used to predict $N_2O$ emissions[61], was correlated with $N_2O$ emissions ($R^2 = 0.16$; $p = 0.001$; Supplementary Fig. 1) but was not significant in a model that contained $NO_3^-$. Agricultural intensity explained 25% of the variability in $N_2O$ fluxes (log GAM $R^2 = 0.25$; $p < 0.001$), but again was not significant in a model containing $NO_3^-$ and VWC as proximal controllers of $N_2O$ emission. The effect of agriculture on $N_2O$ emissions was mainly related to cultivation (Fig. 2b). We could detect no significant difference between $N_2O$ emissions from agriculturally unused sites and pastures or hay fields. Thus non-agricultural sources of elevated N (e.g. from chronically elevated atmospheric N deposition), and lower soil water content (e.g. reductions in precipitation) would likely have a similar impact on $N_2O$ emissions as agricultural fertilisation and drainage.

## Discussion

This is the first time that simple, robust global models of $N_2O$ emissions driven by nitrate, moisture and temperature of organic soils have been identified. It is notable that the models encompass temperate, continental, and tropical biomes. Our findings provide more accurate models of the drivers of $N_2O$ emissions from organic soils across a wide range of biomes and management

regimes than heretofore developed. This highlights the importance of soil nitrate, moisture, and temperature in organic soils as significant global contributors to climate change and stratospheric ozone depletion. Our global-scale models show that constantly high soil moisture results in low $N_2O$ emissions, whereas drainage creates fluctuation around the intermediate soil moisture and thus increases $N_2O$ emissions from organic soils. The temperature effect on $N_2O$ emissions emphasises the importance of considering the warm fertile soils in the global $N_2O$ budget. The implication of this work is that wetland conservation should be a priority for climate change mitigation, particularly given the evidence for future increases in the magnitude and frequency of summer droughts[60]. The anticipated large $N_2O$ emissions from N-rich drained organic soils can be mitigated through wetland conservation and restoration, and through appropriate soil management, such as reduced tillage, nutrient management and improved crop rotations[46]. These have been implemented to some extent in developed countries but need to be further expanded and extended, as a matter of urgency, to tropical and sub-tropical regions.

## Methods

**Study sites**. Our global soil- and gas-sampling campaign was conducted during the vegetation periods between August 2011 and March 2017, following a standard protocol. We sampled 58 organic-soil sites using criteria for organic soils (>12% soil carbon content in the upper 0.1 m) adapted from the FAO World Reference Base for Soils[62] in 23 regions throughout the A (rainy tropical), C (temperate), and D (boreal) climates of the Köppen classification (Fig. 1; Supplementary Data 1). Both natural and artificially drained sites were identified, based on the proximity of drainage ditches, water table height, and characteristic vegetation. The hydrology and trophic status of the natural sites ranged from groundwater-fed swamps and fens to ombrotrophic peat bogs. We measured the most important environmental drivers that were possible.

**Field and laboratory measurements**. Within each region, we established sites to capture the full range of environmental conditions as described above. The depth of the topsoil organic horizon ranged from 0.1 to 6 m across the sites. Land use ranged from natural mire and swamp forest to managed grassland and arable land. A four-grade agricultural-intensity index was used to quantify the effect of land conversion: 0—no agricultural land use (natural mire, swamp, or bog forest), 1—moderate grazing or mowing (once a year or less), 2—intensive grazing or mowing (more than once a year), and 3—arable land (directly fertilised or unfertilised). The agricultural intensity index was estimated based on observation and contacts with site managers and local researchers.

At each site, 1 to 4 stations were established 15–500 m apart to maximise the environmental variance. Each station was instrumented with 3–5 white opaque PVC 65 L truncated conical chambers 1.5–5 m apart and a 1-m-deep observation well (a 50-mm-diametre perforated PP-HT pipe wrapped in geotextile). The total number of chambers was 444. $N_2O$ fluxes were measured using the static chamber method[63] using PVC collars of 0.5 m diameter and 0.1 m depth installed in the soil. A stabilisation period of 3–12 h was allowed before gas sampling to reduce the disturbance effect on fluxes from inserting the collars. The chambers were placed into water-filled rings on top of the collars. Gas was sampled from the chamber headspace into a 50 mL glass vial every 20 min during a 1-h session. The vials had been evacuated in the laboratory 2–6 days before the sampling. At least three sampling sessions per location were conducted over 3 days. The gas samples were brought to the University of Tartu and analysed for $N_2O$ concentration within 2 weeks using two Shimadzu GC-2014 gas chromatographs equipped with ECD, TCD, and Loftfield-type autosamplers[63]. $N_2O$ fluxes were determined on the basis of linear regressions obtained from consecutive $N_2O$ concentrations in three to five samples taken when the chamber was closed, resulting in 61 negative and 502 positive $N_2O$ fluxes ($p < 0.05$ for the goodness of fit to linear regression). There were 982 additional insignificant fluxes ($p > 0.05$) below the gas-chromatography measuring accuracy (regression change of $N_2O$ concentration, δv, <10 ppb) reported as zero in the database and included in the analyses.

Water-table height was recorded daily from the observation wells during the gas sampling at least 8 h after placement. Soil temperature was measured at 10, 20, 30, and 40 cm depth. Soil samples of 150–200 g were collected from the chambers at 0–10 cm depth after the final gas sampling. Humification was rated on the von Post scale, 1 to 10 grades from completely undecomposed to completely decomposed peat[64]. The soil samples were brought to Estonian University of Life Sciences for chemical and physical analyses. During transport, the samples were kept below the ambient soil temperature at which they were collected.

In the laboratory, plant-available phosphorus (P) was determined on a FiaStar5000 flow-injection analyser (KCl extractable). Plant-available potassium (K) was determined from the same solution by the flame-photometric method, and plant-available magnesium (Mg) was determined from a 100-mL $NH_4$-acetate solution with a titanium-yellow reagent on the flow-injection analyser. Available calcium (Ca) was analysed using the same solution by the flame-photometrical method. Soil pH was determined on a 1 N KCl solution[65]. Soil $NH_4$-N and $NO_3$-N were determined on a 2 M KCl extract of soil by flow-injection analysis[65]. Total nitrogen and carbon contents of oven-dry samples were determined using a dry-combustion method on a varioMAX CNS elemental analyser. The organic-matter content of oven-dry soil (SOM) was determined by loss on ignition at 360 °C. We determined gravimetric water content (GWC) as the difference between the fresh and oven-dry weight divided by the oven-dry weight[66]. Bulk density was determined as follows[67]:

$$BD = (D_{bm} \cdot D_{bo})/(SOM \cdot D_{bm} + (1 - SOM) \cdot D_{bo}), \quad (3)$$

where:

BD is bulk density, g cm$^{-3}$,

$D_{bm}$ is the empirically determined bulk density of the mineral fraction (2.65 g cm$^{-3}$)[66],

$D_{bo}$ is the empirically determined bulk density of the organic fraction (0.035–0.23 g cm$^{-3}$ according to the von Post humification scale[68]), and

SOM is the organic content of the oven-dry soil, g g$^{-1}$.

We determined VWC as[66]:

$$VWC = GWC \cdot BD, \quad (4)$$

where:

VWC is volumetric water content, m$^3$ m$^{-3}$,

GWC is gravimetric water content, Mg Mg$^{-1}$, and

BD is bulk density, Mg m$^{-3}$.

For normalising the soil temperature to possible local optima we divided our soil-temperature measurements with the mean air temperature at the nearest weather station in the warmest month of the year[69] (KNMI Climate Explorer http://climexp.knmi.nl; Supplementary Data 1).

**Statistical analysis.** Principal component analysis (PCA), Spearman's rank correlation and stepwise multiple regression of site-mean efflux vs. the environmental parameters were used. The tests were run using both untransformed and log-transformed $N_2O$ fluxes. Before the log-transformation, a constant value was added to all fluxes to account for negative values. Normality of the variables and the residuals was checked by the Shapiro–Wilk test. Neither the $N_2O$ fluxes nor their logarithms were normally distributed ($p < 0.05$); this is a commonly reported issue with $N_2O$. Therefore only a nonparametric test such as Spearman's rank correlation and generalised additive models (GAM) could be applied. We used the mgcv package of the R Project to calculate the GAM regressions using minimal smoothness ($k = 3$). We reported $p$-values (significance level $p < 0.05$) from the summaries of the GAM regressions produced by the summary.gam package of the R Project. We only reported GAM regressions when the residuals were normally distributed. As a presumption for the stepwise multiple regression, the independent variables were checked for GAM concurvity—we only reported multiple relationships with a variance inflation factor <10 between the independent variables. We tested the presence of a boundary in our data[59]. The test compared the density of points in the region of the data set's upper envelope to the expected density of the upper envelope of a bivariate normally distributed data set of the same size[59].

**Literature analysis.** In order to compare our model with independent external data, we surveyed literature referenced in the Thomson Reuters Web of Science. The search terms were: $N_2O$ and organic soil and nitrous oxide and organic soil. We only included publications that reported time series of at least a year's duration that reported $N_2O$ fluxes and simultaneous soil temperature and soil moisture observations (either VWC or WFPS). Eleven papers[17,24,31,32,34,53–58] qualified under these criteria. The study sites were fairly evenly distributed throughout major organic soil regions of the world. Only three of these papers reported soil $NO_3^-$ concentrations[17,24,58]. We converted the WFPS values to VWC as follows[66]:

$$VWC = WFPS \cdot TP, \quad (5)$$

where:

VWC is volumetric water content, m$^3$ water m$^{-3}$ fresh soil,

WFPS is water-filled porosity, m$^3$ water m$^{-3}$ pore space, and

TP is total porosity, m$^3$ pore space m$^{-3}$ soil.

To standardise the highly different absolute $N_2O$ values among data sets we normalised them by scaling to the maximum value measured at each site[70]. We calculated average relative $N_2O$ fluxes in 15 soil temperature classes: 0 °C to 2 °C, 2 °C to 4 °C, … and 28 °C to 30 °C, and 10 soil moisture classes: 0% to 10%, 10% to 20%,… and 90% to 100%. Linear and GAM regressions with minimal smoothness ($k = 3$) were determined between soil temperature, soil moisture and both the individual and average relative $N_2O$ fluxes.

**Data availability.** The data reported in this paper are deposited in the PANGAEA repository https://doi.pangaea.de/10.1594/PANGAEA.885897.

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

## Acknowledgements

This study was supported by the Ministry of Education and Science of Estonia (the SF0180127s08 grant), the Estonian Research Council (the IUT2-16, IUT2-17 and PUTJD619 grants); and the EU through the European Regional Development Fund (ENVIRON and EcolChange Centres of Excellence, Estonia), the 7[th] Framework People programme (the PIRSES-GA-2009-269227 grant), the European Social Fund (Doctoral School of Earth Sciences and Ecology, Estonia), the IAEA's Coordinated Research Project D12010, and Labex VOLTAIRE (ANR-10-LABX-100-01). We are sincerely grateful to the assistance of Dr. I. Filippov, G. Gabiri, Dr. J. B. Gallagher, I. Gheorghe, Dr. W. Hartman, Dr. R. Iturraspe, C.K. Luswata, S. Mander, Dr. M. Metspalu, R. Moreton, Dr. H. Óskarsson, Dr. J. Paal, Dr. E.S.-O. Parrodi, Dr. S. Pellerin, Dr. S. Pihu, K. Raudsepp, Dr. F. Sabater, D. Silveira Batista, and Dr. E. Uuemaa in study-site selection and field investigation. Our work benefitted from technical assistance from Dr. C. Vohla, discussions with Dr. T. Leppelt and Dr. A. Kanal, and a pre-review by Prof. U. Skiba. Dr. T. Ligi, Dr. M. Metspalu, K. Oopkaup and Dr. M. Truu contributed to the perspective microbiological study.

## Author contributions

Ü.M. conceived the study and planned the field campaign. J.P. managed the field campaign in most regions. S.E., J.Jä., K.K., A.K., E.D.L., M.M., T.P., J-.O.S., K.Soh., K.Soo., A.T., G.V., and Ü.M. performed the field work. S.U., F.L.-D., E.D.L., M.M.T., W.J.M., F.S., K.St., J.A.V., and S.S.Z. provided local expertise in site selection and interpretation. A.T. measured the gas samples. J.P., Ü.M., K.Soh. and A.A. compiled and analysed the data, and created the map in Fig. 1. M.E. and J.T. performed the principal

component analysis. J.T.A.V., K.B.B., N.B.D., S.U., J.Ja., L.K., A.K., F.L.-D., E.D.L., A.L., K.L., W.J.M., C.M., Ü.N., B.O., F.S., and J.A.V. made suggestions for the analyses and the paper. J.P. and Ü.M interpreted the results and wrote the paper.

## Additional information

**Competing interests:** The authors declare no competing interests.

