## [Peer Review File · Nature Communications]

Reviewers' comments:

Reviewer #1 (Remarks to the Author):

This manuscript presents results from 64 studies sites across the world with organic soils which were each visited once during the period 2011-2017 to make a dozen or more N₂O chamber flux measurements over the course of a few days. The compiled results show that the strongest predictors of flux were extractable soil nitrate, followed by soil volumetric water content (VWC), with little effect of temperature and other factors. An empirical model with these two parameters explained 69% of the variation (nitrate alone explained 64%). This empirical model also performed well on data from similar sites already published in three literature studies. The results support a conceptual model and several empirical studies on predominantly mineral soils cited by the authors, in which a combination of N availability (represented by extractable nitrate in this case) and VWC are used as predictors of NO, N₂O, or N₂ emissions from soils. At intermediate VWC values, N₂O is the dominant gaseous product, which is consistent with the finding in this manuscript of a quadratic fit with highest emissions around 50% VWC. Higher water contents presumably inhibit nitrification, and thus limit nitrate substrate for denitrification, and also cause N₂ to become the dominant end product of denitrification. These relationships have been demonstrated extensively in mineral soils of grassland, forest, and agricultural ecosystems, but wetlands with organic soils have been less well studied.

Temperature was shown to be a poor predictor of N₂O emissions in this study. Although the authors briefly acknowledge that temperature affects denitrification rates when all other factors are held constant, I fear that they are not adequately representing the limitations of their study design. I understand that it would not have been feasible to study seasonal variation of temperature and N₂O emissions at so many sites, but I think it should be acknowledged that their conclusion of a limited effect of temperature is based on the assumption that they can substitute space for time – i.e., they analyze the effect of temperature across sites rather than across seasons within a site. Nor can one conclude that rising temperatures at a site as the result of ongoing and future climate change would not affect N₂O emissions either through a direct temperature effect on denitrifying activity or through an indirect effect of temperature on evapotranspiration, water balance, and then soil moisture.

There is a similar problem with the use of only a snapshot of nitrate concentrations in the soil measured during their brief visit to each site. Nitrate is often very dynamic, responding to both seasonal patterns of climate and plant and microbial activity and also rapid responses to synoptic weather patterns such as rainfall events. This concern about nitrate dynamics would be a good justification for showing the relationship between N₂O emissions and the C:N ratio of the soil. The authors mention that nitrate and C:N are negatively correlated across sites and that both are correlated (with opposite signs) with N₂O, but they chose to use nitrate rather than C:N ratio in their

model because nitrate performed somewhat better as a predictor of N₂O. However, dynamic nitrate concentrations could differ widely at a site depending on what day someone measures it, and so a prediction based on the model presented in this manuscript may vary widely depending on when nitrate is measured at a site. In contrast, soil C:N is relatively stable, and when it changes, it usually takes years to detect a change. There are probably more available datasets on soil C:N than on soil nitrate, so the former may be a more useful driver of predictive models, both due to availability of data and relative stability of the driving variable. The present study is justified on the need for improved understanding of how climate change and land use change will affect N₂O emissions from organic soils, but for the present model to be useful, one would first have to predict how climate change and land use change will affect nitrate production. That, in turn, is likely related, at least in part, to soil C:N. It is fine to emphasize nitrate as the superior performer in this dataset, but I think it would be useful and important to present the other regression results and other empirical model fits, including using C:N, in a supplemental file.

The authors should point out in their discussion section that they sampled each site for only a few days and that they visited temperate and boreal sites during the growing seasons. Therefore, their study was not designed to detect the effect of seasonal or synoptic scale variation of temperature, soil nitrate, and other factors within each site.

The final sentence of the abstract (“Thus our simple model predicts the impact of land use or climate change on N₂O emission across a wide range of habitats”) is not supported by the data, because the model uses nitrate as a predictor but does not predict how land use and climate affect nitrate. The results support the inference that wetland conservation would probably help avoid large N₂O losses by avoiding conditions that favor nitrate production, but it would be helpful to the reader to provide literature references that demonstrate nitrate accumulation when wetlands are drained and/or plowed.

Reviewer #2 (Remarks to the Author):

The project that has led to the production of this paper has evidently involved a very wide international collaboration, and the organisers are to be complimented on their achievement. The authors are proposing a very simple, and therefore appealing, model that predicts N₂O emissions from organic soils on the basis of only two variables: soil nitrate and volumetric water content. However, as I make clear below, the absence of temperature as a controlling factor is of concern.

The Introduction is clear and informative, and adequately referenced, and most of the Methods section is satisfactory. But I have several specific questions/comments relating to this section:

- a) Why is the criterion for categorising a soil as organic a C content of 10% (line 103)? My reading of Ref 50 is a threshold of 12% (which also accords with the citation of the same source in the IPCC 2006 Guidelines).
- b) Given that the 10% threshold is used here, why are data points included where the value is <10%?
- c) If a soil is 100% organic, that implies a %C content of 50-58%, depending on the different conversion values common in the literature. How, then, are some of the values in the table explained – for example the group of samples from the French Pyrenees site with 65-81%C (lines 615-638 in the table)?
- d) Lines 125-135: Presumably the 50 ml glass vials into which the air samples were transferred were evacuated. If so the text should say so, and also indicate whether this was done immediately prior to filling the vials, or whether evacuated vials were sent from afar to be used at the sites. There is a potential issue of quality control here. Also, it says in lines 134-135: “At least three sessions per location were run within three days.” Is my understanding correct that this 3-day period was the only emission measurement period at any one site?
- e) I cannot claim to be familiar with the statistical techniques used, and must leave evaluation of the statistics section to other reviewers.

My other area of concern is in the Results & Discussion. It relates to the two sentences beginning on line 247, which state: “Only one site (‘Tasmania arable’) received fertiliser whereas the nitrate in the other 63 sites originated from natural sources such as nitrification, atmospheric deposition, runoff and groundwater. The IPCC emission factors from fertiliser application rates and land-use would not provide a result in such ecosystems.” (my emphasis). I cannot agree with this statement, at least for the agricultural sites studied. There are special IPCC default emission values (“EF2” in IPCC Guidelines 2006) for organic soils. For tropical organic soils the value is 16 kg N₂O-N/ha/year, range 5-48 kg. There are three tropical sites with intensive land use in Table S1: the Malaysian oil palm plantation, and the Myanmar and Uganda arable. Given the relatively constant temperatures throughout the year for such environments, it is interesting to see what the effect would be of simply scaling up the 3-day N₂O emissions snapshot to a 365-day period, for these sites, to see how the results would compare with IPCC predictions.

For the Malaysian site, temp = 27-30°, the mean emission rate is 294.3 ug N₂O-N/m²/hr = 70.6 g N₂O-N /ha/day = 25.8 kg N₂O-N /ha/year;

For Myanmar, 14-19° (upland) the figures are 125.5 ug N₂O-N/m²/hr = 30.1 g/ha/day = 11.0 kg/ha/year.

For Uganda, 17-20° (upland) the figures are 498 ug N₂O-N/m²/hr = 119.8 g/ha/day = 43.7 kg/ha/year.

So the “annual fluxes” obtained by this simplistic upscaling all fall within the IPCC tropical default range.

The other relevant point here is that the fluxes at these 3 sites are much greater than elsewhere, and is it not the case that evidence for temperature being a key factor in determining emissions is being lost because of the preponderance of sites with lower temperatures in this large dataset? Should there not be some stratification of the data into temperature zones, rather than lumping all the data points into a single analysis?

Minor point: labelling of the N₂O flux axis in Fig 4 should run from bottom to top, not top to bottom.

Reviewer #3 (Remarks to the Author):

Review of "Soil moisture as the main global driver of nitrous oxide emission from nitrogen-rich organic soils"

Given the complexities of biogeochemical cycling in soils, it is indeed useful to determine the main drivers of nitrous oxide emissions for a subset of soils. The authors investigate the relationships between nitrous oxide emissions and a range of site parameters measured at a number of organic soil sites across the globe. By normalising some measure of water content (claimed to be volumetric water content) at each site, a relationship with log N₂O emissions was found, and in combination with NO₃⁻ concentration a global model of N₂O emissions could explain much of the variance in the measured N₂O emissions. The few independent studies of N₂O emissions from organic soils found in the literature allowed a limited comparison, rather than a test of the model as claimed in the manuscript. Interestingly, no relationship between soil temperature and N₂O emissions was found, although it is not clear whether thermal adaptation, which was considered in the introduction, was taken into account in the analysis. The strongest predictor of N₂O emissions across the field sites was soil NO₃⁻, however the title of the manuscript seems to suggest that soil moisture was the main global driver.

There are a number of technical and mechanistic issues which I feel have not been adequately dealt with in the manuscript.

Firstly, there needs to be a recognition that metrics such as water content are an indirect way to capture the influence of a number of properties on a number of processes. Whilst it is pragmatic to

use an empirical approach such as that used in this paper, it is also important to recognise the complexities and to be transparent about the underlying processes. In relation to water content we are trying to gain information about redox potential and also the ability of a number of gases and solutes to move in the soil matrix. In particular, we are interested in the reactants and products of a range of biogeochemical reactions whose diffusivity are differentially impacted on not only by the volume of soil occupied by gas and solutes, but also the size, connectivity and tortuosity of soil pores. A number of papers have attempted to deal with these issues and although there may not yet be a perfect answer on what is the best metric to use, and there may never, it is critical that we are very clear and careful about how we measure soil moisture.

In contrast, the measurement and subsequent use of a soil moisture metric in this manuscript was not clear. To accurately measure volumetric water content, the bulk density needs to be estimated. The measurement of bulk density is not trivial, and significant errors can be made if it is not done carefully. The usual way in which bulk density of soil is measured is by very carefully taking a number of in situ soil cores of known volume, drying the contents, weighing the contents, and using the particle density to calculate the volume of the solids. For mineral soils, a particle density of around 2.65 g m⁻³ is often assumed. For organic soils the particle density may vary and may need to be determined. In this manuscript it was stated that the bulk density was determined in the laboratory by filling a cylinder with each soil as close to its field compaction as possible. There was no indication of how this field compaction was determined nor a comparison of this largely undefined method to the more standard or robust way of measuring bulk density. All things considered, the actual soil moisture metric used was probably not the volumetric water content. One has to also acknowledge that the bulk density of soil can change over time, so it is not practical to retrospectively measure it properly. The only action I can recommend is to repeat the analysis using gravimetric water content.

In addition to the issues around estimating bulk density and the ability to actually report volumetric water content, it appears that the estimate of volumetric water used was normalised anyway. That may actually assist in dealing with the issue above, in that the normalised gravimetric water content could be the same or very similar to the normalised volumetric water content, depending on how the normalisation was done. Unfortunately the methods section was not detailed enough to enable me to determine this with confidence. Given the ongoing interest in the relationships between N₂O fluxes with soil moisture metrics (a number of the cited papers deal with it, and a number of subsequent papers have since been published) it would be an important finding if indeed the normalised gravimetric water content was a useful predictor of N₂O emissions in organic soils, so I encourage the authors to revisit this issue.

The published data provide some comparison of the relationship between N₂O fluxes and moisture at an annual time scale, however the normalisation and the presentation of the data on different scales limits the usefulness of the comparison. There was no indication of how episodic N₂O fluxes are in organic soils, so questions remain over the validity of comparing annual data to instantaneous data. Although the analysis suggests that there may be some agreement that fluxes are highest at

intermediate soil moisture contents, it is a stretch to claim that there was an independent test of the model developed from the sampled sites.

The authors suggest that drainage of wet organic soils, or wetting up of drained organic soils can dramatically increase N₂O emissions. Is that solely based on extrapolation from the model, or did any of the field sites demonstrate this, noting that water table depth was measured?

The authors acknowledge that a number of other factors could be important. It is interesting that no relationship between soil temperature and N₂O emissions was found. Was any attempt made to normalise the temperature data in a similar way to how the soil moisture data were handled? C/N was found to be more related to N₂O fluxes than C alone. Would some measure of labile carbon have been and even better predictor? Could redox potential have been measured using probes orthogonal to the observation well?

What quality control was done on the submitted data? On eyeballing the data there seem to be some pH values and water level values that do not make sense. There may be other issues that I missed.

Reviewers' comments:

Reviewer #1 (Remarks to the Author):

This manuscript presents results from 64 studies sites across the world with organic soils which were each visited once during the period 2011-2017 to make a dozen or more N₂O chamber flux measurements over the course of a few days. The compiled results show that the strongest predictors of flux were extractable soil nitrate, followed by soil volumetric water content (VWC), with little effect of temperature and other factors. An empirical model with these two parameters explained 69% of the variation (nitrate alone explained 64%). This empirical model also performed well on data from similar sites already published in three literature studies. The results support a conceptual model and several empirical studies on predominantly mineral soils cited by the authors, in which a combination of N availability (represented by extractable nitrate in this case) and VWC are used as predictors of NO, N₂O, or N₂ emissions from soils. At intermediate VWC values, N₂O is the dominant gaseous product, which is consistent with the finding in this manuscript of a quadratic fit with highest emissions around 50% VWC. Higher water contents presumably inhibit nitrification, and thus limit nitrate substrate for denitrification, and also cause N₂ to become the dominant end product of denitrification. These relationships have been demonstrated extensively in mineral soils of grassland, forest, and agricultural ecosystems, but wetlands with organic soils have been less well studied.

Thank you for a very comprehensive digest of our paper.

Temperature was shown to be a poor predictor of N₂O emissions in this study. Although the authors briefly acknowledge that temperature affects denitrification rates when all other factors are held constant, I fear that they are not adequately representing the limitations of their study design. I understand that it would not have been feasible to study seasonal variation of temperature and N₂O emissions at so many sites, but I think it should be acknowledged that their conclusion of a limited effect of temperature is based on the assumption that they can substitute space for time – i.e., they analyze the effect of temperature across sites rather than across seasons within a site. Nor can one conclude that rising temperatures at a site as the result of ongoing and future climate change would not affect N₂O emissions either through a direct temperature effect on denitrifying

activity or through an indirect effect of temperature on evapotranspiration, water balance, and then soil moisture.

We fully agree with the reasoning for the temperature effect. We now acknowledge the limited time span of our measurements per site (LL 276–277) and present a multiple-regression model that includes soil temperature (LL 279–285).

There is a similar problem with the use of only a snapshot of nitrate concentrations in the soil measured during their brief visit to each site. Nitrate is often very dynamic, responding to both seasonal patterns of climate and plant and microbial activity and also rapid responses to synoptic weather patterns such as rainfall events. This concern about nitrate dynamics would be a good justification for showing the relationship between N₂O emissions and the C:N ratio of the soil. The authors mention that nitrate and C:N are negatively correlated across sites and that both are correlated (with opposite signs) with N₂O, but they chose to use nitrate rather than C:N ratio in their model because nitrate performed somewhat better as a predictor of N₂O.

Indeed LogC:N predicted 16% of LogN₂O emissions ($R^2=0.16$) while LogNO₃⁻ predicted 63% of LogN₂O emissions.

However, dynamic nitrate concentrations could differ widely at a site depending on what day someone measures it, and so a prediction based on the model presented in this manuscript may vary widely depending on when nitrate is measured at a site. In contrast, soil C:N is relatively stable, and when it changes, it usually takes years to detect a change. There are probably more available datasets on soil C:N than on soil nitrate, so the former may be a more useful driver of predictive models, both due to availability of data and relative stability of the driving variable. The present study is justified on the need for improved understanding of how climate change and land use change will affect N₂O emissions from organic soils, but for the present model to be useful, one would first have to predict how climate change and land use change will affect nitrate production. That, in turn, is likely related, at least in part, to soil C:N. It is fine to emphasize nitrate as the superior performer in this dataset, but I think it would be useful and important to present the other regression results and other empirical model fits, including using C:N, in a supplemental file.

Again, we agree with the reviewer's criticism and added a remark about the missing relationship into the text. In particular, we have included the relationship with the C:N ratio in Fig. S1.

The authors should point out in their discussion section that they sampled each site for only a few days and that they visited temperate and boreal sites during the growing seasons. Therefore, their study was not designed to detect the effect of seasonal or synoptic scale variation of temperature, soil nitrate, and other factors within each site.

Agreed. We added the statement: "This may have been partially due to the short time span of our measurements per site" to LL 276–277 and the paragraph: "Because we sampled each site for only a few days and that we visited temperate and boreal sites during the growing seasons this study was not designed to detect the effect of seasonal or synoptic scale variation of temperature, soil nitrate, and other factors within each site. Thus our global models are only applicable to estimate daily N₂O emissions based on instantaneous environmental conditions at organic-soil sites. Annual-average N₂O emissions at sites under a seasonal climate may be more difficult to draw from our model. Yet the model could be useful to estimate N₂O emissions at sites under a lack of seasonal variation in environmental conditions such as the tropical humid climate." to LL 286–293.

The final sentence of the abstract ("Thus our simple model predicts the impact of land use or climate change on N₂O emission across a wide range of habitats") is not supported by the data,

because the model uses nitrate as a predictor but does not predict how land use and climate affect nitrate.

Agreed. We removed the problematic sentence from the abstract.

The results support the inference that wetland conservation would probably help avoid large N₂O losses by avoiding conditions that favor nitrate production, but it would be helpful to the reader to provide literature references that demonstrate nitrate accumulation when wetlands are drained and/or plowed.

Agreed. We added the requested references to L 261.

Reviewer #2 (Remarks to the Author):

The project that has led to the production of this paper has evidently involved a very wide international collaboration, and the organisers are to be complimented on their achievement. The authors are proposing a very simple, and therefore appealing, model that predicts N₂O emissions from organic soils on the basis of only two variables: soil nitrate and volumetric water content.

Thank you for the positive words.

However, as I make clear below, the absence of temperature as a controlling factor is of concern. The Introduction is clear and informative, and adequately referenced, and most of the Methods section is satisfactory. But I have several specific questions/comments relating to this section:

a) Why is the criterion for categorising a soil as organic a C content of 10% (line 103)? My reading of Ref 50 is a threshold of 12% (which also accords with the citation of the same source in the IPCC 2006 Guidelines).

b) Given that the 10% threshold is used here, why are data points included where the value is <10%?

Accepted. In our initial version, we adapted the FAO definition of organic soils in our database to be inclusive of all peaty soils. However, we fully realise that our decision was arbitrary. In the current version we strictly followed the FAO threshold (12% soil carbon) and included only sites above that. It resulted in a loss of three regions but owing to the relatively small loss of data, the main relationships remained the same and, in some cases, the correlation even improved.

c) If a soil is 100% organic, that implies a %C content of 50-58%, depending on the different conversion values common in the literature. How, then, are some of the values in the table explained – for example the group of samples from the French Pyrenees site with 65-81%C (lines 615-638 in the table)?

We re-entered and double checked all values in the database. The C contents in fact go up to 66%.

d) Lines 125-135: Presumably the 50 ml glass vials into which the air samples were transferred were evacuated. If so the text should say so, and also indicate whether this was done immediately prior to filling the vials, or whether evacuated vials were sent from afar to be used at the sites. There is a potential issue of quality control here.

We clarified this in L 133 as follows: "The vials had been evacuated in the laboratory 2–6 days before the sampling." We have carried out tests at the lab by keeping the evacuated vials unused for weeks, without noticeable air leaks. Another clear validation criterion

was the linear rise in the GC analyses – all samples with R values below 0.95 ($p < 0.05$) were omitted. This is mentioned in the text.

Also, it says in lines 134-135: "At least three sessions per location were run within three days." Is my understanding correct that this 3-day period was the only emission measurement period at any one site?

Yes, that is correct. We added the statement: "This may have been partially due to the short time span of our measurements per site" to LL 276–277 and the paragraph: "Because we sampled each site for only a few days and that we visited temperate and boreal sites during the growing seasons this study was not designed to detect the effect of seasonal or synoptic scale variation of temperature, soil nitrate, and other factors within each site. Thus our global models are only applicable to estimate daily N₂O emissions based on instantaneous environmental conditions at organic-soil sites. Annual-average N₂O emissions at sites under a seasonal climate may be more difficult to draw from our model. Yet the model could be useful to estimate N₂O emissions at sites under a lack of seasonal variation in environmental conditions such as the tropical humid climate." to LL 286–293.

e) I cannot claim to be familiar with the statistical techniques used, and must leave evaluation of the statistics section to other reviewers.

My other area of concern is in the Results & Discussion. It relates to the two sentences beginning on line 247, which state: "Only one site ('Tasmania arable') received fertiliser whereas the nitrate in the other 63 sites originated from natural sources such as nitrification, atmospheric deposition, runoff and groundwater. The IPCC emission factors from fertiliser application rates and land-use would not provide a result in such ecosystems." (my emphasis). I cannot agree with this statement, at least for the agricultural sites studied. There are special IPCC default emission values ("EF2" in IPCC Guidelines 2006) for organic soils. For tropical organic soils the value is 16 kg N₂O-N/ha/year, range 5-48 kg. There are three tropical sites with intensive land use in Table S1: the Malaysian oil palm plantation, and the Myanmar and Uganda arable. Given the relatively constant temperatures throughout the year for such environments, it is interesting to see what the effect would be of simply scaling up the 3-day N₂O emissions snapshot to a 365-day period, for these sites, to see how the results would compare with IPCC predictions.

For the Malaysian site, temp = 27-30°, the mean emission rate is 294.3 ug N₂O-N/m²/hr = 70.6 g N₂O-N /ha/day = 25.8 kg N₂O-N /ha/year;

For Myanmar, 14-19° (upland) the figures are 125.5 ug N₂O-N/m²/hr = 30.1 g/ha/day = 11.0 kg/ha/year.

For Uganda, 17-20° (upland) the figures are 498 ug N₂O-N/m²/hr = 119.8 g/ha/day = 43.7 kg/ha/year.

So the "annual fluxes" obtained by this simplistic upscaling all fall within the IPCC tropical default range.

Thank you very much for showing the validity of IPCC guidelines in the context of our results. That is indeed a relevant point and our statement was rather speculative. We replaced the sentence with the paragraph: "Upscaling our three tropical sites with intensive land use (the Malaysian oil palm plantation, and the Myanmar and Uganda arable; Table S1) to a year's duration and comparing them with the special default emission values ("EF2") in IPCC Guidelines 2006 for tropical organic soils⁶⁶ (16 kg N₂O-N ha⁻¹ y⁻¹, range 5–48 kg ha⁻¹ y⁻¹) gave us the following results. For the Malaysian site, soil temperature was 27–30°, the mean emission rate was 294.3 ug N₂O-N m⁻² h⁻¹ = 25.8 kg N₂O-N ha⁻¹ y⁻¹. For Myanmar, 14–19° (upland) the figures were 125.5 ug N₂O-N m⁻² h⁻¹ = 11.0 kg N₂O-N ha⁻¹ y⁻¹. For Uganda, 17–20° (upland) the figures were 507.3 ug

$\text{N}_2\text{O-N m}^{-2} \text{ h}^{-1} = 44.5 \text{ kg/ha/year}$. Thus the “annual fluxes” obtained by this simplistic upscaling all fell within the IPCC tropical default range.” in LL 293–302.

The other relevant point here is that the fluxes at these 3 sites are much greater than elsewhere, and is it not the case that evidence for temperature being a key factor in determining emissions is being lost because of the preponderance of sites with lower temperatures in this large dataset? Should there not be some stratification of the data into temperature zones, rather than lumping all the data points into a single analysis?

We added Table S2 that distinguishes our N_2O -flux measurements between climate zones, and added a sentence “The difference between N_2O emissions from drained and natural sites was clear in all three major climate types (Table S2)”. This shows that regardless of the temperature climate, drainage has an immediate effect on N_2O emissions. Still the best relationship we detected was the one across the whole dataset. However, we fully agree with the reasoning for the temperature effect. We now acknowledge the limited time span of our measurements per site (LL 276–277) and have included a multiple-regression model that incorporates soil temperature (LL 279–285).

Minor point: labelling of the N_2O flux axis in Fig 4 should run from bottom to top, not top to bottom.

We changed the direction of the label.

Reviewer #3 (Remarks to the Author):

Review of “Soil moisture as the main global driver of nitrous oxide emission from nitrogen-rich organic soils”

Given the complexities of biogeochemical cycling in soils, it is indeed useful to determine the main drivers of nitrous oxide emissions for a subset of soils. The authors investigate the relationships between nitrous oxide emissions and a range of site parameters measured at a number of organic soil sites across the globe. By normalising some measure of water content (claimed to be volumetric water content) at each site, a relationship with log N_2O emissions was found, and in combination with NO_3^- concentration a global model of N_2O emissions could explain much of the variance in the measured N_2O emissions. The few independent studies of N_2O emissions from organic soils found in the literature allowed a limited comparison, rather than a test of the model as claimed in the manuscript.

Thank you for explanation. We changed the word ‘test’ to ‘compare’ and ‘comparison’ in LL 195 and 245.

Interestingly, no relationship between soil temperature and N_2O emissions was found, although it is not clear whether thermal adaption, which was considered in the introduction, was taken into account in the analysis.

Indeed, in our current database, log N_2O emissions show a significant but weak relationship with soil temperature alone (40cm depth, $R^2=0.20$; $p<0.001$). However, in a multiple-regression GAM model with Log NO_3^- , the two variables explain 69% of the variation in log N_2O emissions (LL 279–285).

The strongest predictor of N_2O emissions across the field sites was soil NO_3^- , however the title of the manuscript seems to suggest that soil moisture was the main global driver.

Agreed. We changed the title accordingly to "Nitrogen-rich organic soils under warm, well-drained conditions: global hotspots of nitrous oxide emission".

There are a number of technical and mechanistic issues which I feel have not been adequately dealt with in the manuscript.

Firstly, there needs to be a recognition that metrics such as water content are an indirect way to capture the influence of a number of properties on a number of processes. Whilst it is pragmatic to use an empirical approach such as that used in this paper, it is also important to recognise the complexities and to be transparent about the underlying processes. In relation to water content we are trying to gain information about redox potential and also the ability of a number of gases and solutes to move in the soil matrix. In particular, we are interested in the reactants and products of a range of biogeochemical reactions whose diffusivity are differentially impacted on not only by the volume of soil occupied by gas and solutes, but also the size, connectivity and tortuosity of soil pores. A number of papers have attempted to deal with these issues and although there may not yet be a perfect answer on what is the best metric to use, and there may never, it is critical that we are very clear and careful about how we measure soil moisture.

We totally agree this is the common procedure in long-term field studies. Our objective was to explain N₂O emissions as a function of potential key physical and chemical factors measured either in the field or from a soil sample bag. More sophisticated soil parameters may be mechanistically more correct but also may be impossible to obtain for global upscaling.

In contrast, the measurement and subsequent use of a soil moisture metric in this manuscript was not clear.

Agreed. We changed the soil moisture metric (see the explanation three paragraphs below).

To accurately measure volumetric water content, the bulk density needs to be estimated. The measurement of bulk density is not trivial, and significant errors can be made if it is not done carefully. The usual way in which bulk density of soil is measured is by very carefully taking a number of in situ soil cores of known volume, drying the contents, weighing the contents, and using the particle density to calculate the volume of the solids. For mineral soils, a particle density of around 2.65 g m⁻³ is often assumed. For organic soils the particle density may vary and may need to be determined.

In the current version of our manuscript we applied the 2.65 g cm⁻³ for the mineral matter proportion of the soil and 0.5 g cm⁻³ for the organic matter proportion (Helmke, P.A. *The Chemical Composition of Soils. In Handbook of Soil Science (2000)*).

In this manuscript it was stated that the bulk density was determined in the laboratory by filling a cylinder with each soil as close to its field compaction as possible. There was no indication of how this field compaction was determined nor a comparison of this largely undefined method to the more standard or robust way of measuring bulk density. All things considered, the actual soil moisture metric used was probably not the volumetric water content. One has to also acknowledge that the bulk density of soil can change over time, so it is not practical to retrospectively measure it properly. The only action I can recommend is to repeat the analysis using gravimetric water content.

Agreed. We followed your recommendation (see the explanation four paragraphs below).

In addition to the issues around estimating bulk density and the ability to actually report volumetric water content, it appears that the estimate of volumetric water used was normalised anyway. That may actually assist in dealing with the issue above, in that the normalised gravimetric water

content could be the same or very similar to the normalised volumetric water content, depending on how the normalisation was done.

Agreed. This is a correct assumption.

Unfortunately the methods section was not detailed enough to enable me to determine this with confidence. Given the ongoing interest in the relationships between N₂O fluxes with soil moisture metrics (a number of the cited papers deal with it, and a number of subsequent papers have since been published) it would be an important finding if indeed the normalised gravimetric water content was a useful predictor of N₂O emissions in organic soils, so I encourage the authors to revisit this issue.

Thank you once again for the valuable comment. We abandoned our calculation of VWC and instead normalised our GWC ($\text{g}=\text{cm}^3$ of water g^{-1}) with the average particle densities of mineral (2.65 g cm^{-3}) and organic matter (0.5 g cm^{-3}) in the respective proportions of mineral and organic matter in fresh soil (LL 160–178). The resulting NGWC (normalised GWC) values are in Table S1. We compared the NGWC values against intact soil cores in polypropylene cylinders (237cm^3) collected at the Colombia, French Guiana, Kyrgyzstan, Myanmar and Tierra del Fuego sites, which had been transported to the laboratory among the rest of the soil samples, weighed, oven-dried, weighed again and used for calculating the volumetric water content (VWC) of the core from the difference. The match was 63% ($\text{VWC} = 0.86\text{NGWC} + 0.0087$; $R^2=0.63$; $n=105$; $p<0.01$; see the figure below). The relationship between N₂O emission and NGWC is analysed in LL 230–236.

Figure. VWC from intact soil cores vs. normalised GWC. $\text{VWC} = 0.86\text{NGWC} + 0.0087$; $R^2=0.63$; $n=105$; $p<0.01$

The published data provide some comparison of the relationship between N₂O fluxes and moisture at an annual time scale, however the normalisation and the presentation of the data on different scales limits the usefulness of the comparison. There was no indication of how episodic N₂O fluxes are in organic soils, so questions remain over the validity of comparing annual data to instantaneous data. Although the analysis suggests that there may be some agreement that fluxes are highest at intermediate soil moisture contents, it is a stretch to claim that there was an independent test of the model developed from the sampled sites.

We agree that high N₂O fluxes are very episodic. Unfortunately a comparison with the published annual time series (LL 245–251) is the most solid way to support the discussion on that issue. On top of that we can only very broadly discuss this issue even at an annual time scale, because N₂O flux estimates are typically made using chamber-based determinations of instantaneous fluxes, such as in our measurements.

In addition we added a paragraph: "Because we sampled each site for only a few days and that we visited temperate and boreal sites during the growing seasons this study was not designed to detect the effect of seasonal or synoptic scale variation of

temperature, soil nitrate, and other factors within each site. Thus our global models are only applicable to estimate daily N₂O emissions based on instantaneous environmental conditions at organic-soil sites. Annual-average N₂O emissions at sites under a seasonal climate may be more difficult to draw from our model. Yet the model could be useful to estimate N₂O emissions at sites under a lack of seasonal variation in environmental conditions such as the tropical humid climate. Upscaling our three tropical sites with intensive land use (the Malaysian oil palm plantation, and the Myanmar and Uganda arable; Table S1) to a year's duration and comparing them with the special default emission values ("EF2") in IPCC Guidelines 2006 for tropical organic soils⁶⁶ (16 kg N₂O-N ha⁻¹ y⁻¹, range 5–48 kg ha⁻¹ y⁻¹) gave us the following results. For the Malaysian site, soil temperature was 27–30°, the mean emission rate was 294.3 ug N₂O-N m⁻² h⁻¹ = 25.8 kg N₂O-N ha⁻¹ y⁻¹. For Myanmar, 14–19° (upland) the figures were 125.5 ug N₂O-N m⁻² h⁻¹ = 11.0 kg N₂O-N ha⁻¹ y⁻¹. For Uganda, 17–20° (upland) the figures were 507.3 ug N₂O-N m⁻² h⁻¹ = 44.5 kg/ha/year. Thus the "annual fluxes" obtained by this simplistic upscaling all fell within the IPCC tropical default range." in LL 286–302.

The authors suggest that drainage of wet organic soils, or wetting up of drained organic soils can dramatically increase N₂O emissions. Is that solely based on extrapolation from the model, or did any of the field sites demonstrate this, noting that water table depth was measured?

That conclusion is indeed solely based on our model. Unfortunately the short time span of measurements per site did not permit a significant demonstration of this based on field data.

The authors acknowledge that a number of other factors could be important. It is interesting that no relationship between soil temperature and N₂O emissions was found.

In our current database, log N₂O emission shows a weak relationship with soil temperature alone (40cm depth R²=0.20). However, in a multiple-regression GAM model with Log NO₃, the two variables explain 69% of variation in log N₂O emissions (LL 279–285).

Was any attempt made to normalise the temperature data in a similar way to how the soil moisture data were handled?

To calculate VWC in the old version, GWC was normalised with the weight of 1L of fresh soil measured at the soil laboratory. We are not sure what an appropriate normalisation factor would be for temperature.

C/N was found to be more related to N₂O fluxes than C alone. Would some measure of labile carbon have been and even better predictor? Could redox potential have been measured using probes orthogonal to the observation well?

Both the proposed parameters could potentially be useful in mechanistically explaining N₂O fluxes. We even plan to analyse our soil samples currently stored under –18°C for carbon fractions. Nonetheless, for a global extrapolation, the availability of sophisticated measurements, such as carbon fractions or soil ORP, is probably scarce. Hence, for a global understanding of N₂O emissions it is important to relate N₂O emissions with more widely used parameters and factors derived from them.

What quality control was done on the submitted data? On eyeballing the data there seem to be some pH values and water level values that do not make sense. There may be other issues that I missed.

These values were not a result of poor quality control but just copy-paste errors. We did not have any pH data on the Borneo sites during the preparation of the old version. Fortunately we did not use the false values in any analyses. We re-entered and double checked all values in the current database (Table S1).

Reviewers' comments:

Reviewer #1 (Remarks to the Author):

The authors have adequately addressed most of the concerns that I raised in my previous review. However, in the process of converting their soil moisture measurements to a normalized index, they have introduced an important error, which may be a new error or it may have been a hidden one in their previous use of soil moisture. On lines 159-160 they define gravimetric water content (GWC) as “the difference between the fresh and oven-dry weight of a soil sample divided by the fresh weight.” This is incorrect. Any introductory soils text or lab manual (e.g., <https://lter.kbs.msu.edu/protocols/24>) will explain that GWC should be expressed on a dry-weight basis: “the difference between the fresh and oven-dry weight of a soil sample divided by the oven-dry weight.” The denominator of GWC must be on a dry weight basis so that as water content changes, only the numerator of this ratio changes. One can also see the necessity of expressing GWC on a dry weight basis when combining it with bulk density measurements, such as those used in line 167, which are also always expressed as “g dry soil/cm³ soil space.” The equations on line 162 and 167 are invalid because they mix GWC on a wet-weight basis with particle density which is on a dry weight basis.

I also question the calculation of NC in line 167, because the units do not come out right. Dividing OM and MM by their respective particle densities gives the desired units of cm³/g dry soil, but then GWC is added, which has units of gH₂O/gwetsoil. Even if the latter were converted to the correct representation of GWC (gdrysoil), one would still be adding cm³/g and g/g, which is not possible. There must be something else wrong with this equation.

Another problem is that it isn't explained how MM is estimated. Is it: $MM = 1 - OM$?

I suspect that the incorrect usage of GWC will require that all of the calculations and statistical analyses be redone.

Perhaps a better approach than the proposed NGWC would be to estimate bulk density (BD) from organic matter (OM) content using one of the more appropriate literature-based relationships:

<http://www.nrcresearchpress.com/doi/abs/10.4141/CJSS06008#.WgNUps anG70>

https://www.researchgate.net/publication/232151800_Carbon_Organic_Matter_and_Bulk_Density_Relationships_in_A_Forested_Spodsol

http://www.scielo.br/scielo.php?pid=S0100-06832017000100309&script=sci_arttext

and there are several others in the published and gray literature.

Once you have BD estimated, GWC can be converted to VWC:

$$\text{GWC (gH}_2\text{O/gdrysoil)} \times \text{BD (gdrysoil/cm}^3\text{soil)} = \text{VWC (gH}_2\text{O/cm}^3\text{soil)} = \text{VWC (cm}^3\text{H}_2\text{O/cm}^3\text{soil)}$$

I think that this would be superior to inventing a new term: “normalized gravimetric water content (NGWC).” The authors compare their NGWC to measured VWC in intact cores as a form of validation, so they are clearly trying to estimate VWC. As long as they are transparent about how they estimate BD from measurements of OM, then there shouldn’t be a problem of referring to their soil moisture estimates a VWC.

For the sake of comparing to literature where water filled pore space (WFPS) is reported, they could also convert their data a similar way once they have estimates of BD and VWC:

Total porosity (TP) = 1 – (BD/PD), where a weighted average of particle density (PD) is calculated:

$$\text{PD} = [(\text{OM} \times 0.5) + (\text{MM} \times 2.65)] / (\text{OM} + \text{MM}) \text{ (assuming that } \text{OM} + \text{MM} = 1)$$

$$\text{WFPS} = \text{VWC} / \text{TP}$$

It is curious that the authors equate VWC and WFPS in Fig. 5, which is also incorrect. VWC is $\text{cm}^3\text{H}_2\text{O/cm}^3\text{totalsoil}$ space, whereas WFPS is $\text{cm}^3\text{H}_2\text{O/cm}^3\text{soilpore}$ space. These values may be similar in highly organic soils, because the BD is low and a very large fraction of the total soil space is made up of voids (pores), which can be filled with either air or water. Therefore, in plots like Fig. 5, the difference between plotted values of VWC and WFPS may not be very different along the X axis for organic soils, but VWC is always at least somewhat less than WFPS and they are not identical. I fear that the current presentation could confuse students and make them think that these expressions of soil moisture are equivalent. The difference becomes very important when the soil is dominated by minerals rather than OM. For example, 50% VWC would be nearly 100% WFPS for most mineral soils, where TP is seldom above 50-60%. Given that the soils in the present study have a range of OM values, down to 12%, the difference between VWC and WFPS could become important for some of the soils at the lower end of that range.

I am concerned that there isn’t enough clarification that wetland drainage can cause decomposition, which releases inorganic-N, especially at sites where the OM has a low C:N ratio. As a result, I have trouble with the concluding statements on lines 309-313:

“conversion of wetlands to pastures or hay fields has little direct effect on N₂O emissions. Non-agricultural sources of elevated N (e.g. from chronically elevated atmospheric N deposition), and lower soil water content (e.g. reductions in precipitation) would likely have a similar impact on N₂O emissions as agricultural fertilisation and drainage.”

The sites converted to pastures or hay field probably already lost the N from the organic matter that was decomposed when the wetlands were first drained. There may no longer be elevated nitrate or N₂O emissions once the pasture or hayfield has been established for many years, but the statement implies that there would be no increase in nitrate or N₂O emissions during the conversion process. There is no evidence to support that statement and I would guess that it is not true with respect to the transition (conversion) period.

N deposition from the atmosphere (non-ag sources) could have similar impacts as fertilisation or drainage if the deposition rates were high enough to sufficiently elevate the soil nitrate (e.g., above the 5 mg/g hotspot threshold reported in this study), which is unlikely in rural areas of the Arctic, Congo, or Amazon, but could happen in wetlands downwind of industrial and urban sources.

Reviewer #2 (Remarks to the Author):

I welcome the readiness of the authors to revise their manuscript in the light of the comments made by myself and the other referees on the original submission. The revisions alter the emphasis and conclusions quite significantly, and provide a new perspective on the impact of elevated temperatures characteristic of tropical soils on N₂O emissions. I am recommending to the editors that the paper be accepted.

Reviewer #3 (Remarks to the Author):

Thank you for considering and incorporating many of the reviewer comments into you revised manuscript. The other two reviewers also made some excellent points.

The one outstanding comment from my review relates to normalising the temperature response. I encourage you to read the section on the response of N₂O emissions to temperature in my paper, "Concepts in modelling N₂O emissions from land use" (Plant and Soil 2008 309:147-167). You will find references to a number of papers, including the use of adaptive temperature functions in models (see for example, Parton et al 2001 Geophys Res 106:17403-17419; and Breuer and Butterbach-Bahl 2005 Australian Journal of Soil Research 43:689-694). These and subsequent papers may give you some ideas on how to better account for the temperature response across sites.

Based on climate data for each of your sites, you may be able to normalise a temperature optimum to some site specific value such as the mean temperature in the warmest month of the year, for example. Doing so may allow you to better account for the contribution of temperature to N₂O emissions, which is something the other reviewers were also concerned about. It is possible that by allowing for adaptive temperature functions, the contribution of temperature to instantaneous N₂O fluxes will become more transparent, which may indeed change the model, and hence the focus of the entire manuscript.

Ryan Farquharson

Reviewer #1 (Remarks to the Author):

The authors have adequately addressed most of the concerns that I raised in my previous review. However, in the process of converting their soil moisture measurements to a normalized index, they have introduced an important error, which may be a new error or it may have been a hidden one in their previous use of soil moisture. On lines 159-160 they define gravimetric water content (GWC) as “the difference between the fresh and oven-dry weight of a soil sample divided by the fresh weight.” This is incorrect. Any introduction soils text or lab manual (e.g., <https://lter.kbs.msu.edu/protocols/24>) will explain that GWC should be expressed on a dry-weight basis: “the difference between the fresh and oven-dry weight of a soil sample divided by the oven-dry weight.” The denominator of GWC must be on a dry weight basis so that as water content changes, only the numerator of this ratio changes. One can also see the necessity of expressing GWC on a dry weight basis when combining it with bulk density measurements, such as those used in line 167, which are also always expressed a “g dry soil/cm³ soil space.” The equations on line 162 and 167 are invalid because they mix GWC on a wet-weight basis with particle density which is on a dry weight basis.

Thank you for pointing out the conceptual error in our calculations. We recalculated our GWC values on a dry basis, according to the suggestion by Reviewer #1, as the difference between the fresh and dry weight divided by the dry weight (highlighted in green in lines 161–163).

I also question the calculation of NC in line 167, because the units do not come out right. Dividing OM and MM by their respective particle densities gives the desired units of cm³/g dry soil, but then GWC is added, which has units of gH₂O/gwetsoil. Even if the latter were converted to the correct representation of GWC (gdrysoil), one would still be adding cm³/g and g/g, which is not possible. There must be something else wrong with this equation.

Another problem is that it isn't explained how MM is estimated. Is it: $MM = 1 - OM$?

I suspect that the incorrect usage of GWC will require that all of the calculations and statistical analyses be redone.

Perhaps a better approach than the proposed NGWC would be to estimate bulk density (BD) from organic matter (OM) content using one of the more appropriate literature-base relationships: <http://www.nrcresearchpress.com/doi/abs/10.4141/CJSS06008#.WgNUpsanG70>
https://www.researchgate.net/publication/232151800_Carbon_Organic_Matter_and_Bulk_Density_Relationships_in_A_Forested_Spodosol
http://www.scielo.br/scielo.php?pid=S0100-06832017000100309&script=sci_arttext
and there are several others in the published and gray literature.

Once you have BD estimated, GWC can be converted to VWC:
 $GWC (gH_2O/gdrysoil) \times BD(gdrysoil/cm^3soil) = VWC (gH_2O/cm^3soil) = VWC (cm^3H_2O/cm^3soil)$

I think that this would be superior to inventing a new term: “normalized gravimetric water content (NGWC).” The authors compare their NGWC to measured VWC in intact cores as a form of validation, so they are clearly trying to estimate VWC. As long as they are transparent about how they estimate BD from measurements of OM, then there shouldn't be a problem of referring to their soil moisture estimates a VWC.

Thank you enormously for the comprehensive approach to soil water content. We applied it and it worked very well. You will see from the manuscript that calculating BD following Périé C. & Ouimet R. Organic carbon, organic matter and bulk density relationships in boreal forest soils. Can J Soil Sci 88: 315-325 (2008) and the VWC equation suggested by the referee improved our VWC model (Fig. 3b) and increased the multiple-regression model with LogNO₃ and VWC to R²=0.72 (Fig. 4a). Respective changes to text are highlighted in green in lines 163–179 and 239–240.

For the sake of comparing to literature where water filled pore space (WFPS) is reported, they could also convert their data a similar way once they have estimates of BD and VWC:
Total porosity (TP) = 1 - (BD/PD), where a weighted average of particle density (PD) is calculated:
 $PD = [(OM \cdot 0.5) + (MM \cdot 2.65)] / (OM + MM)$ (assuming that $OM + MM = 1$)
 $WFPS = VWC / TP$

It is curious that the authors equate VWC and WFPS in Fig. 5, which is also incorrect. VWC is $\text{cm}^3\text{H}_2\text{O}/\text{cm}^3\text{totalsoilspace}$, whereas WFPS is $\text{cm}^3\text{H}_2\text{O}/\text{cm}^3\text{soilporespace}$. These values may be similar in highly organic soils, because the BD is low and a very large fraction of the total soil space is made up of voids (pores), which can be filled with either air or water. Therefore, in plots like Fig. 5, the difference between plotted values of VWC and WFPS may not be very different along the X axis for organic soils, but VWC is always at least somewhat less than WFPS and they are not identical. I fear that the current presentation could confuse students and make them think that these expressions of soil moisture are equivalent. The difference becomes very important when the soil is dominated by minerals rather than OM. For example, 50% VWC would be nearly 100% WFPS for most mineral soils, where TP is seldom above 50-60%. Given that the soils in the present study have a range of OM values, down to 12%, the difference between VWC and WFPS could become important for some of the soils at the lower end of that range.

Thank you again for offering a solution to a difficult problem. Fortunately all papers that reported only WFPS also disclosed their TP. That made it easy to convert the WFPS to VWC (Equation 3 in lines 203–208). The conversion improved the result considerably (Fig. 5).

I am concerned that there isn't enough clarification that wetland drainage can cause

decomposition, which releases inorganic-N, especially at sites where the OM has a low C:N ratio. As a result, I have trouble with the concluding statements on lines 309-313:

“conversion of wetlands to pastures or hay fields has little direct effect on N₂O emissions. Non-agricultural sources of elevated N (e.g. from chronically elevated atmospheric N deposition), and lower soil water content (e.g. reductions in precipitation) would likely have a similar impact on N₂O emissions as agricultural fertilisation and drainage.”

The sites converted to pastures or hay field probably already lost the N from the organic matter that was decomposed when the wetlands were first drained. There may no longer be elevated nitrate or N₂O emissions once the pasture or hayfield has been established for many years, but the statement implies that there would be no increase in nitrate or N₂O emissions during the conversion process. There is no evidence to support that statement and I would guess that it is not true with respect to the transition (conversion) period.

N deposition from the atmosphere (non-ag sources) could have similar impacts as fertilisation or drainage if the deposition rates were high enough to sufficiently elevate the soil nitrate (e.g., above the 5 mg/g hotspot threshold reported in this study), which is unlikely in rural areas of the Arctic, Congo, or Amazon, but could happen in wetlands downwind of industrial and urban sources.

We agree that here our discussion strayed beyond our evidence. We replaced the passage with: “The effect of agriculture on N₂O emissions was mainly related to cultivation (Fig. 2). We could detect no significant difference between N₂O emissions from agriculturally unused sites and pastures or hay fields.” (lines 314–317).

Minor changes respective to all comments are also highlighted in green.

Reviewer #2 (Remarks to the Author):

I welcome the readiness of the authors to revise their manuscript in the light of the comments made by myself and the other referees on the original submission. The revisions alter the emphasis and conclusions quite significantly, and provide a new perspective on the impact of elevated temperatures characteristic of tropical soils on N₂O emissions. I am recommending to the editors that the paper be accepted.

Thank you very much for your positive review.

Reviewer #3 (Remarks to the Author):

Thank you for considering and incorporating many of the reviewer comments into your revised manuscript. The other two reviewers also made some excellent points.

The one outstanding comment from my review relates to normalising the temperature response. I encourage you to read the section on the response of N₂O emissions to temperature in my paper, "Concepts in modelling N₂O emissions from land use" (Plant and Soil 2008 309:147-167). You will find references to a number of papers, including the use of adaptive temperature functions in models (see for example, Parton et al 2001 Geophys Res 106:17403-17419; and Breuer and Butterbach-Bahl 2005 Australian Journal of Soil Research 43:689-694). These and subsequent papers may give you some ideas on how to better account for the temperature response across sites.

Based on climate data for each of your sites, you may be able to normalise a temperature optimum to some site specific value such as the mean temperature in the warmest month of the year, for example. Doing so may allow you to better account for the contribution of temperature to N₂O emissions, which is something the other reviewers were also concerned about. It is possible that by allowing for adaptive temperature functions, the contribution of temperature to instantaneous N₂O fluxes will become more transparent, which may indeed change the model, and hence the focus of the entire manuscript.

Ryan Farquharson

Thank you for the constructive recommendations. We normalised our soil temperatures to the mean temperature in the warmest month of the year (Parton et al. 2001) from the closest station available at the KNMI Climate Explorer (lines 177–179 highlighted in green). While it looked like a promising approach it did not improve the temperature response ($R^2=0.09$; $p=0.018$; lines 279–281). A particular difference between our data and Parton et al (2001) was the fact that the authors used the normalised temperature function only in the nitrification compartment of their model. It is yet impossible to tell from our data whether the N₂O in our measurements originates from nitrification or denitrification. Another problem with still may have been the short time span of our measurements per site which we have acknowledged as: "This may have been partially due to the short time span of our measurements per site." (lines 281–282). We also tried exponential and the O'Neill regression functions (according to Breuer and Butterbach-Bahl 2005) with our direct measurements but to no improvement on the GAM relationship we already presented (Fig 3c). In addition we ran an analysis separating between our natural and drained sites. It showed that within our drained sites (Table S1; $n=27$) the temperature response was somewhat stronger than across all sites ($R^2=0.27$; $p<0.0078$; Fig. 3c; lines 288–289). This is another piece of evidence to support that under drained conditions, temperature is an important factor of N₂O emissions. That allows us to state in Results and Discussion: "This shows that organic soils exposed to warmer conditions, such as in the tropics, can act as N₂O-emission hotspots where soil moisture is optimal (Fig. 3b) and NO₃⁻ is above a threshold of 5 mg N kg⁻¹ (Fig. 3a)." (lines 289–291), in Conclusions: "The temperature effect on N₂O emissions emphasises the importance of considering the contribution of warm fertile soils to the global N₂O budget." (lines 330–332) and in the final remark of the Abstract: "As soil temperature together with NO₃⁻ explained 69% of N₂O emission, tropical wetlands should be a priority for N₂O management." (lines 58–60).

Respective minor changes are also highlighted in green.

REVIEWERS' COMMENTS:

Reviewer #1 (Remarks to the Author):

The authors have adequately addressed my concerns.

Reviewer #3 (Remarks to the Author):

I am satisfied with how the authors' have addressed the reviewers' comments. Note that some minor edits will be required.

One additional point to consider is that simple regression based approaches such as depicted in figure 3 are often not able to define the true relationship between predictor and response variables when multiple factors are involved. How does the GAM approach compare to approaches that attempt to examine the relationship when other factors are not limiting (e.g. see Milne et al 2005, 2006, 2006 and Lark et al 2004)? I suspect that the dataset is not large enough to enable a boundary line approach, but perhaps there are other statistical methods to define the maximal response.

Response Letter

Dear Reviewer #3,

Thank you for a positive review and another valuable comment (in italics). Please see our point-to-point response below.

Reviewer #3 (Remarks to the Author):

I am satisfied with how the authors' have addressed the reviewers' comments. Note that some minor edits will be required.

Thank you very much!

One additional point to consider is that simple regression based approaches such as depicted in figure 3 are often not able to define the true relationship between predictor and response variables when multiple factors are involved. How does the GAM approach compare to approaches that attempt to examine the relationship when other factors are not limiting (e.g. see Milne et al 2005, 2006, 2006 and Lark et al 2004)? I suspect that the dataset is not large enough to enable a boundary line approach, but perhaps there are other statistical methods to define the maximal response.

That is indeed a useful approach for any environmental response analysis. We tested the presence of a boundary in our temperature data using the procedure of Milne et al (2006). Lines 157–161 in our manuscript present the results of the boundary-line test; lines 315–318 state the methods for the boundary-line test. The test showed that the envelope of the data points does not represent a boundary. This is another piece of evidence suggesting that the high N₂O fluxes were measured in soils where temperature was not the limiting factor.

Kind regards,

Dr. Jaan Pärn
School of Geography, Earth and Environmental Sciences
University of Birmingham

Prof. Ülo Mander
Department of Geography
University of Tartu